# Biomechanics of the Upper Limbs: A Review in the Sports Combat Ambit Highlighting Wearable Sensors

**DOI:** 10.3390/s22134905

**Published:** 2022-06-29

**Authors:** Andrés Blanco Ortega, Jhonatan Isidro Godoy, Dariusz Slawomir Szwedowicz Wasik, Eladio Martínez Rayón, Claudia Cortés García, Héctor Ramón Azcaray Rivera, Fabio Abel Gómez Becerra

**Affiliations:** National Center for Research and Technological Development, Prolongación Av. Palmira s/n esq. Apatzingán, Cuernavaca 62490, Mexico; dariusz.sd@cenidet.tecnm.mx (D.S.S.W.); eladio.mr@cenidet.tecnm.mx (E.M.R.); claudia.cg@cenidet.tecnm.mx (C.C.G.); hector.azcaray10mk@cenidet.edu.mx (H.R.A.R.); fabio.gomez@cenidet.edu.mx (F.A.G.B.)

**Keywords:** biomechanics, inertial sensors, wearable sensors, muscle power, mechanical power, upper extremity

## Abstract

Over time, inertial sensors have become an essential ally in the biomechanical field for current researchers. Their miniaturization coupled with their ever-improvement make them ideal for certain applications such as wireless monitoring or measurement of biomechanical variables. Therefore, in this article, a compendium of their use is presented to obtain biomechanical variables such as velocity, acceleration, and power, with a focus on combat sports such as included box, karate, and Taekwondo, among others. A thorough search has been made through a couple of databases, including MDPI, Elsevier, IEEE Publisher, and Taylor & Francis, to highlight some. Research data not older than 20 years have been collected, tabulated, and classified for interpretation. Finally, this work provides a broad view of the use of wearable devices and demonstrates the importance of using inertial sensors to obtain and complement biomechanical measurements on the upper extremities of the human body.

## 1. Introduction

Biomechanics has arisen from the human need to quantify the interactions of both endogenous and exogenous forces, making use of mechanical laws and postulates. Subsequently, these efforts to understand biological systems and the effects produced by these forces, (applicable to all unicellular or multicellular living beings) have been commonly called biomechanics. [1]. Moreover, biomechanics is firmly based on the physical and mathematical principles developed by human beings throughout history; in which great personalities such as Newton, Descartes, and Euler, among others, have left their mark. [2]. In essence, these biomechanical principles can be applied to bodies at rest or in motion.

Scientists have been faced with the arduous task of approximating the biomechanical behavior of the movements of the human body, arms, feet, and even the torso. Conversely, the interrelationship between strength and movement is important and should be understood as such, setting a precedent for improving the ability and performance in professional athletes or simply preventing possible injuries of these, even for the treatment of musculoskeletal diseases among other applications [3].

According to the International Olympic Committee, boxing, fencing, judo, karate, wrestling, and Taekwondo are considered Olympic combat sports. Generally, in these contact sports, the qualitative and quantitative analyses of the movements are of interest to both the athlete and the coach. A correct analysis influences the execution of a refined technique without loss of energy and with the maximum performance of the athlete. In addition, in terms of medicine, they could even anticipate possible injuries that the athlete may suffer due to muscle overload or malpractice [4].

The movements of the upper extremities generally affect oscillations for the different parts of the human body. Due to this, when trying to measure biomechanical parameters in the torso, head, and even lower extremities employing wireless sensors, they can cause variations. For example, clinical studies [5] report a decrease in the accuracy of the sensors with which respirations are measured as a result of swinging the arms. Furthermore, the reliability of measurements in the rest of the body is compromised if the effects produced by the arms are ignored [6].

This article also addresses an analysis of the existing literature for the development of upper extremity biomechanics with a focus on the field of combat sports. For this purpose, information extracted from research involving the use of inertial sensors to establish a biomechanical model of human movement is included. In addition, interesting insights are presented on how the data were acquired and the processing of the results, reflecting movements of the upper extremities. The article is organized as follows: Section 2 presents biomechanics of interest in the sports field. In Section 3, the reference system of the human body is sketched. Section 4 shows the methodology based in PRISMA protocol. Section 5 exposes the angular displacements of the upper extremities. Section 6 presents some electronic devices as the most prominent measuring instruments in the selected literature. Section 7 deals with an overview of the estimation of joint mechanical power. Section 8 develops the equations that represent joint mechanical power. Finally, Section 9 defines the conclusions of this work, reveals the challenged faced in this research as well as some recommendations for future studies.

## 2. Biomechanics in Sports

Biomechanics has been a fundamental tool since the beginning of the second half of the twentieth century. Since then, it has been considered an autonomous discipline with a wide research outlook. The mental and physical effects have been seen in the Olympic Games [7], due to the constant breaking of world records in athletics (long jump, javelin throw, 100 m flat, and more), boxing, karate, Taekwondo, and others [8]. An example of progress in the performance of an athlete is perfectly palpable by observing the statistics of the world records since the development of biomechanics in the 100 m flat; see Figure 1.

In basketball, several types of research have been reported arguing the proper body position for a 3-point shot, the best exit angle when shooting the ball, and the perfect flexion and extension of the player’s arm/forearm/wrist to achieve the goal of making a basket shot [9], among others. In addition to this, biomechanical analyses have also been reported that thoroughly explain the effects of the forces produced by the player directly after jumping [10], such as those supported by his legs and the ground reaction [11].

Concerning baseball, research focuses on the measurement of the different ways of throwing the ball and the effects of holding the ball with a different style by young, adult, amateur, and professional pitchers. In addition, the recommendations of medical specialists on curveball or fastball throwing practices in youths under 13 years of age are questioned, due to the high stresses placed on the shoulder, joints, and muscles adjacent to the shoulder, which definitively leads to a higher risk of injury [12].

Conversely, based on research focused on combat sports, it can be stated that its priority is to evaluate and analyze the angular positions between the upper and lower limbs.

For this purpose, they take into account the efforts that occur in each limb, forming a database whose principles can be analyzed statistically. Subsequently, the correct technique for a successful stroke can be argued from a biomechanical point of view. This information is useful for athletes, including coaches and doctors, the latter being able to predict possible injuries [13].

Even in sports in full growth, biomechanics is also beginning to be investigated, such as the case of Sanda or San Shou, a variant of freestyle boxing with the use of the lower limbs. The first measurements in force and torque were made, concluding with interesting results. When the speed of muscle contraction increases drastically, muscle strength decreases, indicating an inverse relationship between these two measurements for concentric movements [14].

## 3. Spatial Coordinate System

To obtain a well-established frame of reference and to speak under the same anatomical principles, the coordinate planes on which the human body is delimited must be adequately defined; see Figure 2. The convention shown above for allows the introduction of terms such as flexion, extension, proximal, anterior, abduction, and adduction, among others. Specifically for the upper extremity, it is worth mentioning the movements and the appropriate name according to the plane.

## 4. Methodology

A systematic search was launched, analyzing and classifying an extensive amount of articles in international journals in the area of mechanical engineering and biomechanics. A specific procedure for the search and compilation of the information was carried out and structured for presenting it in this article, which followed the PRISMA framework. The research included in this article comes from different journals. They include the *Journal of Biomechanics* from the Netherlands, *Clinical Biomechanics* from the United Kingdom, and the *Journal of Exercise Science and Fitness* based in Singapore, among others.

The procedure for the search and discrimination of information was carried out in the following stages:Articles published in Spanish and English;Search for articles through keywords such as: biomechanics, upper extremities, biomechanical analysis, biomechanical variables, joint mechanical power, and biomechanical IMU;Evaluate and thoroughly analyze the abstracts of the articles that were selected and make a classification by sport/skill approach;A complete reading of the articles highlighting important results.

Bibliographies from national and international congresses and formal texts on Biomechanics were also used. The classification can be summarized as shown in Figure 3. A total of 154 articles were counted: 29 from IEEE, 25 Taylor & Francis, 16 from Elsevier, and 10 from MDPI among other publishers. To carry out the management and classification, the use of specialized software such as Mendeley has been reported.

## 5. Angular Displacements in the Upper Extremity

When studying upper extremity motion, it is important to delineate between the normal arc of motion for specific joints and the functional arc of rotation from 0 to 150 degrees and pronation and supination from 75 to 85 degrees. The full arc of motion is generally not used for most activities of daily living [15].

The angular movement capacity of the upper extremities can be seen in Table 1, which summarizes the main movements and angles achievable for a physically healthy person, where only for the abduction/adduction movement of the forearm as well as the pronation/supination movement is the view from the frontal plane. For all other movement capabilities, the image presented is from the sagittal plane.

Each joint movement has physical limits that can be described in arcs of motion. The amount of movement expressed in degrees that a joint presents in each of the three planes of space [16] is called the range of motion (ROM). The normal ROM values (according to the Association for the Study of Osteosynthesis) of each of the joints of interest for this research are listed in the fourth column of Table 1.

## 6. Measuring Instruments

The measuring instruments reported in the articles found basically include three: inertial measurement units (IMU’s), electromyographs (EMG) [18] and video recording systems conditioned with high speed cameras. This article focuses on research that has made use of IMUs.

IMUs (inertial measurement units) are complex electronic devices consisting mainly of an accelerometer, gyroscope, and magnetometer. Most of them are built under MEMS/NEMS technology (micro/nano electromechanical systems) [19]. These sensors are used to acquire values proportional to the forces experienced by moving bodies. These specific measurements refer to angular velocity, linear acceleration, and global positioning. For example, in [20], an array of twelve accelerometers placed on a boxing mask was used to evaluate the severity of the blows. The mask was mounted on a Hybrid III test dummy and was conditioned in such a way as to avoid damage to the sensors. Linear acceleration and rotational velocity are captured after a hit with a pendulum at three different speeds. Five areas of the head that are commonly hit in the box were chosen for measurements [21]. Subsequently, a high-pass filter was applied to record the accelerations suffered at different velocities of 3, 5, and 7 m/s, and a linear regression was performed. After plotting the evidence, it was shown that there is a correlation between the severity of the blows (accelerations received in the head) and mild traumatic brain injury (MTBI); see Figure 4b.

In [22], the authors used a wireless inertial sensor attached to the arm to analyze the swing motion of a bowler. A GUI (LabVIEW Guide User Interface) to analyze and visualize the biomechanical parameters critical for a successful throw was developed. They reported the angular velocity of the arm at the maximum possible height during the swing motion and up to the point of the throw, as well as the angle of the wrist relative to the whole arm. Based on the data obtained from the sensor, only the angular velocity Y, roll, and pitch angles were processed and analyzed to derive all other parameters; see Figure 4a.

Some authors use myoelectric signals to investigate biomechanical variables. In this field, [23] proposed a belt placed on the chest as an electrocardiogram. Heart rate measurements were performed on an individual while performing punch-throwing exercises toward a punching bag. Previously, it was instrumented with force sensors and flashing lights, in addition to having the virtue of being able to configure the sequences of these. In this way, the feedback that the individual has to the stimulations of the lights and the speed with which the individual acts accordingly is evaluated and analyzed. Conversely, the lapses of the individual in which their body remains in a state of relaxation, exerting or maintaining the force, is also measured; see Figure 5a.

Conversely, research has been reported that makes use of video-recording technology, using high-speed film cameras, which together with markers positioned on the body of the athletes, are able to follow their trajectory in certain programmed movements. For example, in [24], the kinematics of the arms in field hockey players were analyzed, as well as in the field hockey stick, in addition to measuring with radar the speed with which the puck shoots out. This was for two different types of field hockey stick fastening. In [25], through a projection, the user should respond or not to the attack launched by the projection, with and without prior warning. It was possible to obtain a response from different individuals, both Taekwondo practitioners and others. The variables that were collected were reaction time, maximum peak speed, time from the beginning to the maximum peak speed, average deceleration, precision, and an error constant; see Figure 5b.

These data are shown in Table 2, a general summary of the articles in which common characteristics can be observed in selected articles. These biomechanical data are of great importance, as well as the electronic devices used, the number of test subjects, and even the level of significance for statistical analysis. Likewise, most of the articles express the application of an ethical protocol for human experimentation. Conversely, Table 3 shows a general summary of the inertial sensors that have been used and are reported in the selected articles. This table presents the most important and necessary data that may be of great help to the scientific community.

Despite these efforts and the development of these investigations, there are areas in which some concepts still need to be clearly defined, for example, the estimation of the power used from the upper limb reference or the processes necessary for the calculation of the angles between limbs [26]. These spaces provide opportunities to develop new and innovative research.

During the research, reference values for effective boxing punches can be observed. In [27], the maximum punching force for practitioners ranged from 544 to 1021 N, while their recorded velocity was 6.5 to 9 m/s. However, the fist force for more experienced fighters fluctuated between 3158 and 3242 N, with a speed in the range of 8.1 to 10.2 m/s. These metrics arise from experimentation with three techniques.

Conversely, research has also been expanding in controlled environments using wireless sensors through Universal Software Radio Peripheral (USRP). The monitoring technology comes from the capture of micro-Doppler signals with frequency-modulated continuous-wave (FMCW) radar [28]. These works in which human participation can be individual or multiple focus on the monitoring and recognition of activities of daily life. They are mainly based on the fact that the composition of more than 60% of the water that humans have reflects radio waves. The results are promising due to the ability to provide a health perspective in response to the demand for monitoring activities in people with a disease. Added to these advances the combination of technology with wireless IMU sensors (with which they are used more and more frequently) can revolutionize the perspective of medical diagnoses [29].

## 7. Use and Estimation of the Power Used

When the terms mechanical power and sports power are used in some articles, they can become confused, largely due to the descriptions of athletes and coaches who lack in-depth knowledge of the subject. The mechanical power discussed below is certain to describe the term “power” as an indicator of the consumption/expenditure of mechanical energy of the human body over a period (muscular work) and not, as one would think, an endurance characteristic or performance measurement in an athlete.

Power is a combination of strength and speed, while endurance is a measure of strength; these two cannot be interchanged. Muscle power arises from a metabolic power produced by a chemical process in the human body. Muscle power can produce mechanical power transcribed as the force by the speed of muscle contraction, except for non-conservative power (for example, frictional forces between muscle fibers, heat dissipation, and even when a series of muscles works against each other) and conservative power (in this case, the forces are maintained by elongating muscles).

If power is viewed as the rate at which work is performed or as the amount of energy transferred per unit of time, we can then define it as the relationship between mechanical power, muscular power, and metabolic power, as shown in Figure 6.

Metabolic power can be measured by the rate of oxygen consumption by the human body, from which the energy expenditure by the whole body over time can be estimated. The mechanical power of the human body can be determined by applying the laws of classical mechanics and modeling it as a set of several linked blocks [38,39,40].

## 8. Mechanical Power Estimation Equations

For a better understanding of the behavior of the system, it is necessary to distinguish between joint power, which is the mechanical power generated by the human body on its joints; frictional power losses; kinetic power, which is the rate of change in kinetic energy; gravitational power; and external power, which is the mechanical power of applied external forces and/or moments.

The full power balance equations for the human body are based on a modeling of a chain of N rigid bodies linked together. An individual analysis is performed and then aggregated to obtain the complete balance of the system.

Starting with the translational part, the equation that arises from Newton’s second law applied to the segment “i” of mass $m_{i}$:(1)$$\left(F_{j,i}+F_{G,i}+F_{e,i}-F_{f,i}\right)\cdot v_{i}=m_{i}\cdot a_{i}\cdot v_{i}$$
where $F_{j,i}$ are the forces on the joint, $F_{G,i}$ are the gravitational forces, $F_{e,i}$ are the external forces or applied moments, and finally $F_{f,i}$ are the frictional forces working on the modeled segment. $a_{i}$ and $v_{i}$ represent the linear acceleration and velocity of the segment, respectively. It follows that the translational power balance equation is:(2)$$P_{j,tr,i}+P_{g,tr,i}+P_{e,tr,i}-P_{f,tr,i}=P_{k,tr,i}$$
where, $P_{j,tr,i},\;P_{g,tr,i},\;P_{e,tr,i},\;P_{f,tr,i}$ are the translational joint power, translational gravitational power, translational external power, and translational frictional power, respectively. Finally, $P_{k,tr,i}$ is the translational kinetic power, of each of the above scalar quantities.

To calculate the rotational mechanical power through the Euler equation of motion, expressed in a global reference system and multiplied on both sides by the angular velocity of the segments with the moment of inertia $I_{i}$, we obtain:(3)$$\left(M_{j,i}+M_{e,i}-M_{f,i}\right)\cdot \omega_{i}=\frac{d}{dt}\left(I_{i}\cdot \omega_{i}\right)\cdot \omega_{i}$$
where $M_{j,i}$ is the joint moment, $M_{e,i}$ is the applied external moment, $M_{f,i}$ represents the frictional moment, and $\omega_{i}$ is the angular velocity of the segment. In addition, the rotational power can be set as follows:(4)$$P_{j,ro,i}+P_{e,ro,i}-P_{f,ro,i}=P_{k,ro,i}$$

Therefore, $P_{j,ro,i}$ is equivalent to the rotational joint power, $P_{e,ro,i}$ to the rotational external power, $P_{f,ro,i}$ to the rotational frictional power and y $P_{k,ro,i}$ as the rotational kinetic power all referring to segment “i”. Now, everything makes sense since by joining the translational power with the rotational power of all segments, the constraint forces at the joints do not contribute to the total power equations and are therefore discarded. The total power equation for the system, now written in terms of joint, kinetic, frictional, gravitational, and external power, results as follows:(5)$$P_{j}=P_{k}+P_{f}-P_{g}-P_{e}$$

From the above equation, $\left(P_{j}\right)$ is the joint power, which is directly calculated using the moments at the joint $\left(M_{j}\right)$ and the rotational velocities around the joint $\left(\omega_{j}\right)$.
$$P_{j}=\overset{N-1}{\underset{i=1}{\sum }}M_{j,i+1}\cdot \left(\omega_{i+1}-\omega_{i}\right)=\overset{N-1}{\underset{j=1}{\sum }}M_{j}\cdot \omega_{j}$$

The gravitational power is also set, with g representing gravity, thus:$$P_{G}=\overset{N}{\underset{i=1}{\sum }}v_{i}\cdot m_{i}\cdot g$$
and the power due to friction is constituted by the translational power plus the rotational power, defined as:$$P_{f}=\overset{N}{\underset{i=1}{\sum }}\omega_{i}M_{f,i}+\overset{N}{\underset{i=1}{\sum }}v_{i}F_{f,i}$$
where $F_{f,i}$ characterizes the friction force of the *i*-th segment; in addition, the external power also consists of rotational and translational power to be made up of:$$\;P_{e}=\overset{N}{\underset{i=1}{\sum }}\omega_{i}M_{e,i}+\overset{N}{\underset{i=1}{\sum }}v_{i}F_{e,i}$$
with $F_{e,i}$ as the external force of the “i” segment, and the change of kinetic energy in the segments can be defined as:$$P_{k}=\sum \frac{dE_{seg}}{dt}=\overset{N}{\underset{i=1}{\sum }}\frac{d}{dt}\left(I_{i}\omega_{i}\right)\cdot \omega_{i}+\overset{N}{\underset{i=1}{\sum }}m_{i}a_{i}v_{i}$$

In summary, the mechanical power balance consists of five parts: joint power, kinetic power, gravitational power, external power in the environment, and power due to friction. Joint power is generated by the human body and is the result of muscle power. This implies that a complete estimation of the mechanical power originating from the human body is an interaction between the torques and angular velocities at the joints or indirectly via the algebraic sum of the power as described in the above equation, where usually the latter are approximated depending on the available recording meters, and therefore, not all terms in the power balance equation are estimated to result in a simplified model.

From Figure 7, the forces and moments involved in the upper limb can be appreciated, where the subscripts and superscripts are represented given the following list:
r: rotational;g: gravitational;s: shoulder;to: arm;f: forearm;w: wrist;h: hand.

## 9. Conclusions

In this review, it has been noted that in sports biomechanics, it is of interest to determine the mechanical power for certain movements. This is due to the fact that it is the mechanical principle of the speed at which the athlete works or transfers energy to complete a movement task. Mechanical power is a metric that sports scientists, athletes, and coaches often use for research and training purposes. Estimates of mechanical power are generally limited by the capabilities of motion capture systems, resulting in the need to use simplified power models.

For a complete biomechanical analysis, it is desirable to have a system equipped with high-speed recording cameras, even with infrared capture in such a way that they provide spatiotemporal information for each segment of the human body (videometry). In addition, the use of electronic IMU devices is recommended so that an overview of the state of motion, velocity, and acceleration can be obtained in real time. Finally, do not ignore the complementary data that can be obtained by measuring the myoelectric signals that are sent by the central nervous system through the muscles.

There are several methodologies with which the biomechanics of the upper extremities can be evaluated, including inertial sensors. Although a large amount of data can be obtained with the latter, the adequate use of statistical tools in the processing of these data is necessary. For example, it is necessary to establish the significant limits to consider the experimentation as valid, to establish an expected margin of error, and thus to obtain certainty of the measurements. The applications in which the development of these biomechanical models is focused start from the sports field, medical diagnoses of patients with paraplegic diseases, and even to human–machine interaction through video games.

Another of the challenges little addressed in the development of this work is the fusion of sensors to have more reliable and precise detection. The use of orientation metrics such as Euler angles or quaternions would achieve a deeper insight into the biomechanical processes of combat sport, and athletes can use this information to improve their ability. Sensor fusion is used a lot in artificial intelligence, including mechanical sensors with vision sensors and post-processing techniques to obtain the sensors.

Future work includes the development of wearable sensory systems that do not affect the user’s biomechanics for recording limb movements in combat sports, in such a way that they are resistant to impacts and give signals with minimum error. Probably in the short term will we will be able to interact with a greater number of wireless sensors. In addition, an increase in commonly used devices is predicted, including not only turnkey systems based in IMU technology with inertial sensors, but also blood oxygen, glucose, blood pressure, and EMG sensors, among others.

These portable devices will be assisted with better and more complete applications for total user monitoring, making it possible to bring technology closer to all social strata, which can evidently be enhanced with the booming development of the IoT. The transmission of data from the sensors attached to our bodies in smartwatches, sports bands, and even in clothing opens new opportunities in the field of “big data”. As its use expands, it should require responsible and ethical use of the companies that develop the technology and that manage the data, since the manipulation of these can become a problem due to malicious use. This last assertion may be a topic to be addressed in future research works.

## Figures and Tables

**Figure 1 sensors-22-04905-f001:**
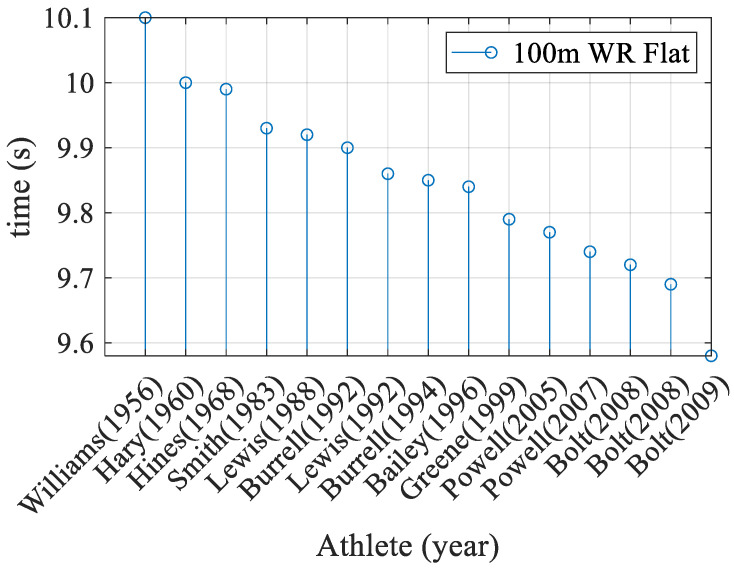
Progress of the 100 m sprint world record.

**Figure 2 sensors-22-04905-f002:**
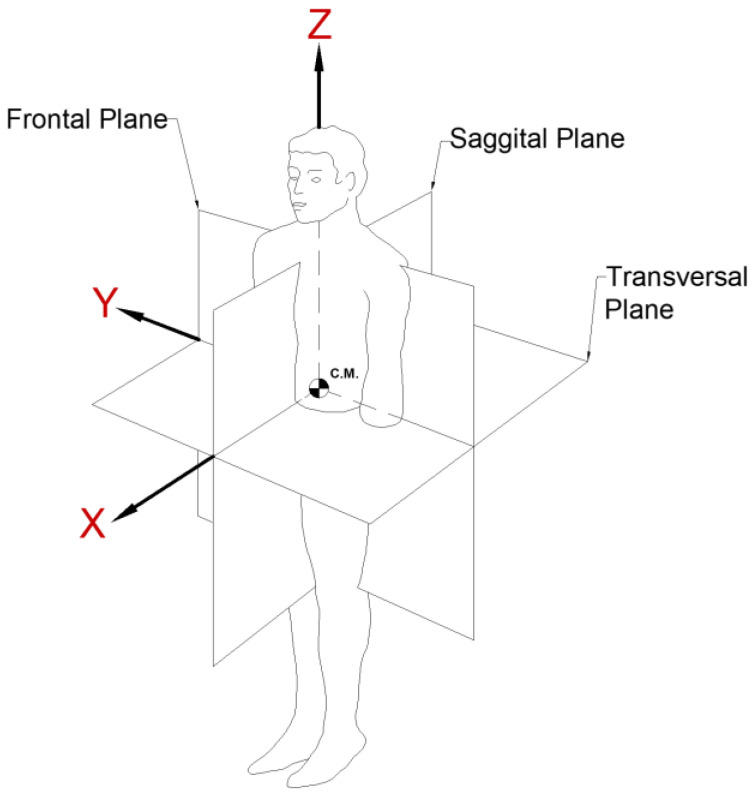
Generalized spatial coordinate system.

**Figure 3 sensors-22-04905-f003:**
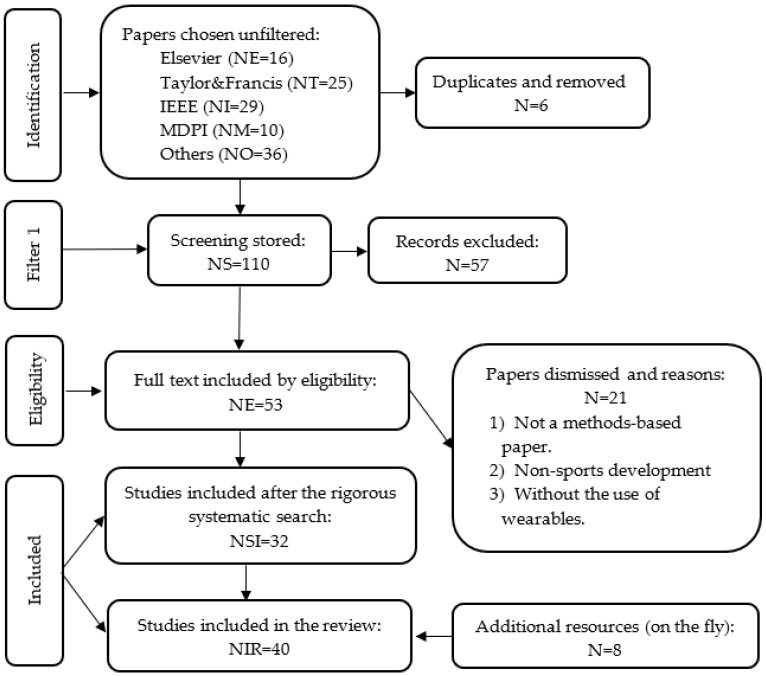
Schematic diagram of material selection.

**Figure 4 sensors-22-04905-f004:**
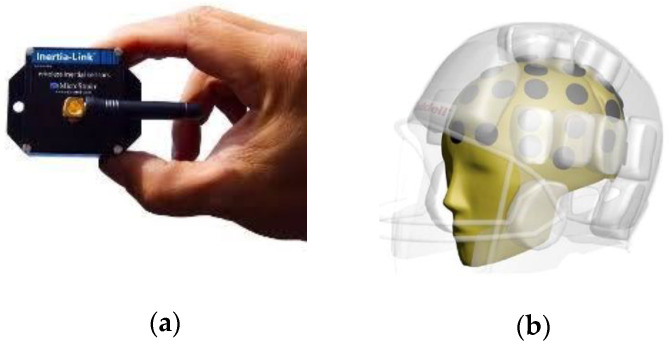
(**a**) Microstrain^®^ wireless inertial sensor; (**b**) Head Impact System with accelerometers.

**Figure 5 sensors-22-04905-f005:**
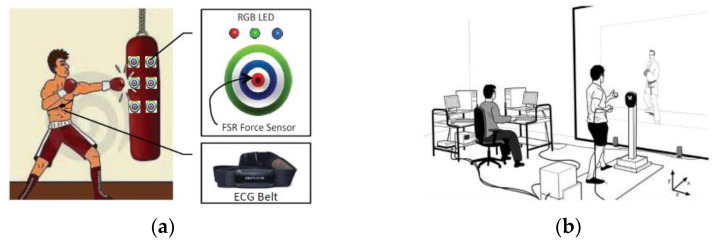
(**a**) Box bag and ECG belt system assembly. (**b**) Pictorial representation of the experimental set.

**Figure 6 sensors-22-04905-f006:**
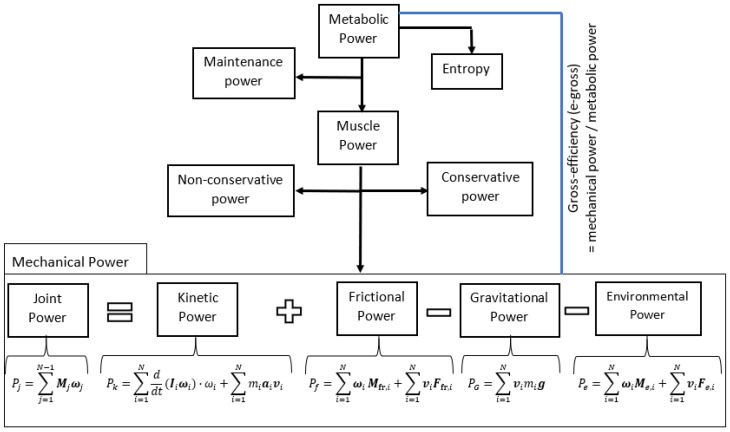
Power flow in human movement.

**Figure 7 sensors-22-04905-f007:**
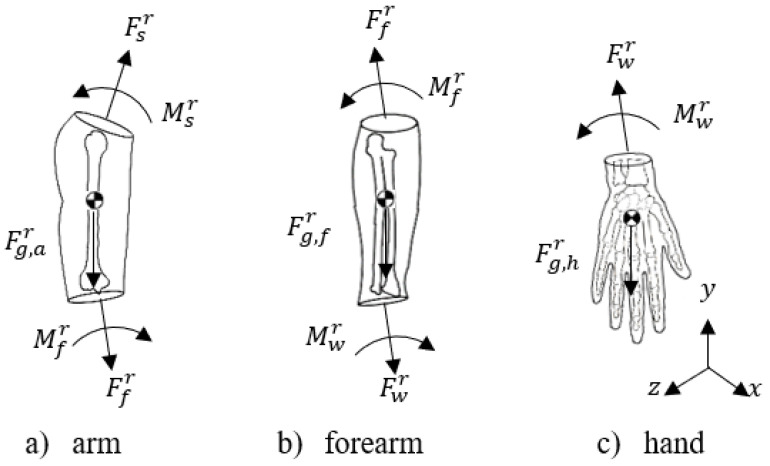
Free-body diagram of a rigid segment model of the right upper limb.

**Table 1 sensors-22-04905-t001:** Normal angular range of motion in the upper extremities [16,17].

Anatomical Reference	Articulation	Movement	Range Min/Max	Illustration
Arm	Shoulder (Glenohumeral)	Flexion	0°/150° to 170°	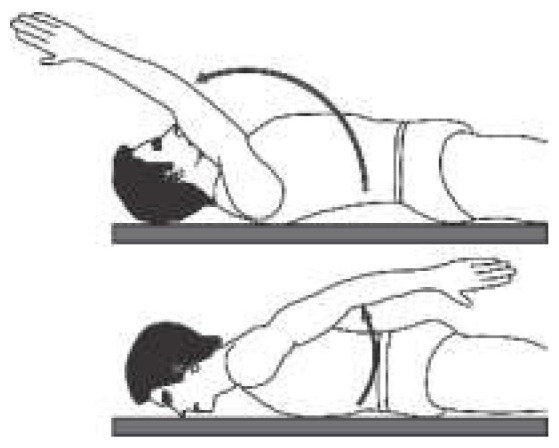
		Extension	0°/40°	
		Abduction	0°/160° to 180°	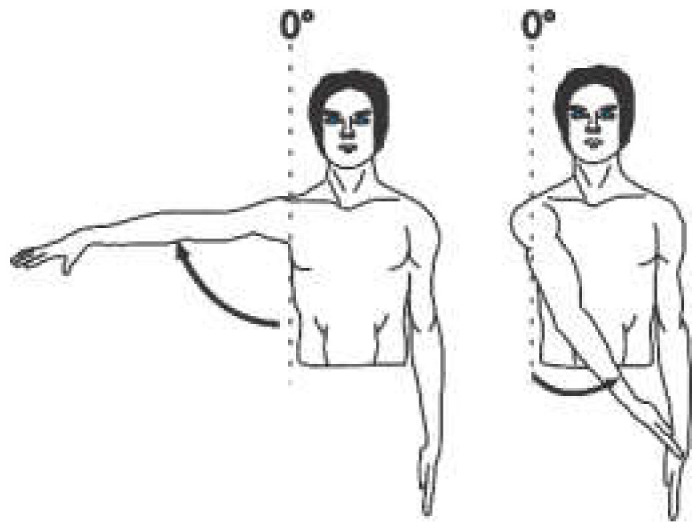
		Adduction	0°/30°	
		External Rotation	0°/70°	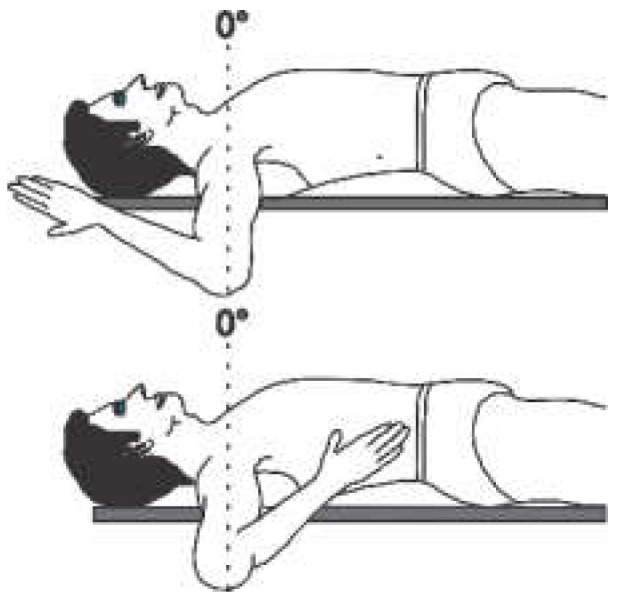
		Internal Rotation	0°/70°	
Forearm	Elbow	Flexion	0°/150°	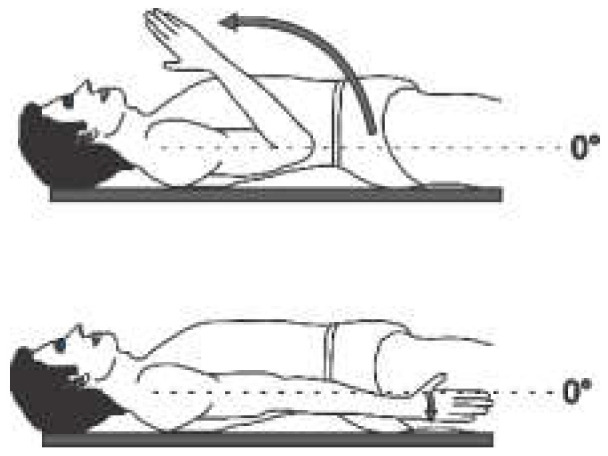
		Extension	0°/10°	
	Proximal and distal radioulnar	Pronation/ Supination	0°/90°	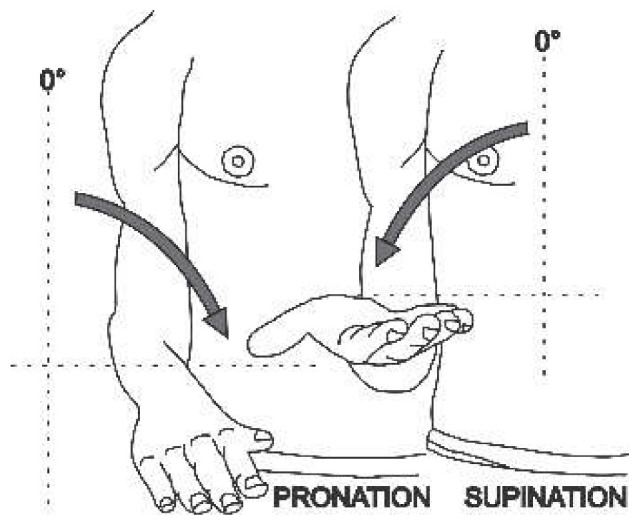

**Table 2 sensors-22-04905-t002:** Most relevant articles, devices used, and methodologies used [13,20,21,23,25,27,30,31,32,33,34,35,36,37].

Article	Sensor/Camera/EMG Used (Software)	DOF	Sport	Velocity Analysis	Acceleration Analysis	Force/Power Analysis	Test Subject
(Thomson, et al., 2013)	Canon MV700 (DartFish TeamPro 4.0)	NA	Box	NA	NA	NA	2
(Beckwith, et al., 2007)	Endevco 7264-B	5 head/neck	Box	No	Rotational and linear	No	U
(Mei, et al., 2014)	Zephyr Bioharness ECG wearable sensor	DA	Box	No	No	No	11
(Martínez de Quel, et al., 2014)	Ascension trakSTAR Model 800 (Cogent 2000 toolbox by Matlab)	DA	Karate	yes	yes	Force	32
(Saponara, 2017)	Sparkfun ADXL377 celda de carga HX711	DA	Sports combat	yes	yes	Force	7
(Chadli, et al., 2014)	Strain gauges, accelerometer	DA	Box	No	yes	Force	11
(Favre, et al., 2015)	3 Accelerometers, 3 gyroscopes	DA	Box	No	No	No	8
(Walilko, et al., 2005)	Tekscan pressure sensor Model 9500	DA	Box	yes	Rotational and linear	Force and power	10
(Shum, et al., 2007)	(MotionGraph)	DA	Box	No	Si	No	2
(Loturco, et al., 2021)	Force Plate AccuPower AMTI	DA	Box	No	No	Force	8
(Dinu and Louis, 2020)	MVN Biomech Link Suit Xsens (Matlab R2010a)	DA	Box	yes	yes	Force	23
(Dinu, et al., 2020)	Suit Xsens (Matlab R2010a)	DA	Box	yes	yes	Force	23
(Mack, et al., 2010)	Endevco 7264-2K, Redlake HG 100K camera (TrackEye Motion analysis)	DA	Box	yes	No	Force	42
(Gavan and Sayers, 2017)	Qualisys Motion Capture System (PowerLab System)	DA	Sports Contact	yes	No	No	24

U: Unspecified. DA: Doesn’t apply.

**Table 3 sensors-22-04905-t003:** Main characteristics of the inertial sensors used in the investigations.

Sensor	Characteristics				
	IMU Type	Accelerometer Accuracy	Gyroscope Accuracy	Sampling Rate	Measurements
Microstrain wireless inertial sensor 3DM-CV5-10	3-DOF Accelerometer and 3DOF gyroscope	±4 g, (optional)	±1000°/s (optional)	Up to 1000 Hz	38 × 24 × 9.7 mm
Vicon Blue Trident	3--DOF Accelerometer 3-DOF Gyroscope 3-DOF Magnetometer	Low-g ±16 g/High-g ±200 g	±2000 deg/s	Up to 1600 Hz	42 × 27 × 11 mm
Sparkfun ADXL377	3-DOF Accelerometer	±200 g	NA	1 kHz	3 × 3 × 1.45 mm
Xsens: Mtw Awinda	3-DOF Accelerometer 3-DOF Gyroscope 3-DOF Magnetometer	± 160 m/s^2^	±2000 deg/s	1000 Hz	148 × 104 × 31.9 mm
Endevco 7264-BM2-300	1-DOF Accelerometer	±500 g	NA	3 kHz	40 × 48 × 18.5 mm
Witmotion WT901BLECL	3-DOF Accelerometer 3-DOF Gyroscope 3-DOF Magnetometer	±160 m/s^2^	±2000 deg/s	200 Hz	51.3 × 36 × 15 mm
InterSense Inertia Cube 3	3-DOF Gyroscope	NA	±2000 deg/s	200 Hz	36.6 × 27.7 × 13.8 mm

## Data Availability

Not applicable.

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
