# Peer review of "Biomechanics of the Upper Limbs: A Review in the Sports Combat Ambit Highlighting Wearable Sensors"

_sensors, 2022, doi:10.3390/s22134905_

Round 1

Reviewer 1 Report

This review paper introduced biomechanics, its applications in sports, and the use of inertial sensors to measure parameters such as velocity, acceleration, and power. As a review paper, this manuscript is not comprehensive enough since it does not include enough literatures and not provide in-depth analysis and discussion. This paper cannot be published on Sensors.

(1)    This paper is more like a brief tutorial of biomechanics rather than a review paper that give comprehensive analysis and discussion of recent progresses made in this field.

(2)    Should provide the rationale for why only research using inertial measurement units was included in this review paper and EMG and video recording systems were ignored.

(3)    It lacks in-depth analysis and discussion on the published works, including the pros, cons, and challenges. There is also no outlook of the future research directions in this field.

(4)    There are only 30 references cited in this manuscript, which is too less for a typical review paper.

Author Response

REVIEWER 1

Comment: This review paper introduced biomechanics, its applications in sports, and the use of inertial sensors to measure parameters such as velocity, acceleration, and power. As a review paper, this manuscript is not comprehensive enough since it does not include enough literatures and not provide in-depth analysis and discussion. This paper cannot be published on Sensors

(1)    This paper is more like a brief tutorial of biomechanics rather than a review paper that give comprehensive analysis and discussion of recent progresses made in this field.

(2)    Should provide the rationale for why only research using inertial measurement units was included in this review paper and EMG and video recording systems were ignored.

(3)    It lacks in-depth analysis and discussion on the published works, including the pros, cons, and challenges. There is also no outlook of the future research directions in this field.

(4)    There are only 30 references cited in this manuscript, which is too less for a typical review paper.

Author’s response: We are grateful to receive your comments. Your assumption is correct, the authors intend to present a summary of the relevant works on upper extremity biomechanics for combat sports. This work contains 3 main pillars, one refers to commonly used biomechanical variables, another to inertial sensors, and finally estimation of mechanical power. Likewise, it is important to note that this work has been reviewed by a couple of people with a command of the English language and who have given the approval of the writing and translation. On the other hand, the number of references that are presented is an indication of the opportunity for this work. There is still much to investigate in the sports area and even more so with the use of inertial sensors that are increasingly developing better capabilities.

1) The information presented in this work is intended to make researchers aware of the progress that biomechanical studies have made with the use of inertial sensors from the beginning of the 21st century to date.

2) Incorporating works that are supported by technologies other than inertial sensors would imply an article that is too long. Based on previous reviews and reviewer suggestions, it was decided to limit the article to only inertial sensors with a focus on combat sports.

3) Some corrections have been made in order to complement this point, in a couple of paragraphs in sections 4, 6, and 9.

4) The number of references only indicates an absence of works specifically on the purpose of this article. As authors, we visualize a gap of opportunity to develop the work.

We welcome your feedback again, major corrections have been made and we look forward to your review again.

Best regards to all reviewers from the authors.

Reviewer 2 Report

Peer-review report 17130309

Thank you for study.

The paper Biomechanics of the upper limbs through wearable sensors: A review in the sports combat ambit, written by authors Andres Blanco Ortega, Jhonatan Isidro Godoy, Dariusz Slawomir Szwedowicz Wasik, Eladio Martinez Rayon, Claudia Cortes Garcia, Hector Ramon Azcaray Rivera, Fabio A. Gomez Becerrase is intended as a review of the use of IMU sensors for upper limb movement analysis.

I appreciate that the authors have attempted to make a commendable effort in collecting and analyzing specific material that has appeared over the last 20 years in order to summarize the use of these sensors.

Major observations:

The way the material is presented, however, is more like a course focused on upper limb biomechanics, with theoretical aspects and anatomical images being mentioned, which should already be known by specialists and researchers who would be interested in the article.

Intended as a review, the material submitted for analysis should follow the PRISMA procedure for analysing bibliographic references. The manuscript should focus more on the methodology for selecting and choosing the articles, and presenting inclusion and exclusion criteria. The article should present extensively the research and biomechanical measurements selected for the purpose of this review.

Therefore, I believe that chapters III (Spatial Coordinate System), IV (Shoulder Musculoskeletal-Articular System) and Table 1 (Normal angular range of motion in upper limbs), a useful table for a biomechanics course, are not necessary in this type of manuscript.

In Chapter V, Methodology, the specific steps of the PRISMA methodology (MDPI recommended methodology for review manuscripts) should be presented.

Chapter IX, Mechanical power estimation equations, is interesting from the point of view of mechanical power estimation, but it is not necessary for this type of manuscript, as it has no bibliographic reference and gives a theoretical presentation of how useful muscle energy is divided during the performance of functional movements of the upper limb.

Chapter X, Conclusions, does not sufficiently detail the applications of IMU sensors and the particularities of the use of these sensors in biomechanical assessments, with references to the materials reviewed in this review manuscript.

Some minor criticisms:

The pdf copy of the manuscript does not contain line numbering and for this reason I cannot make exact references, but the small punctuation errors I noticed will certainly be corrected by the authors when they rewrite the article.

Please note that the title of table 2 is in Spanish.

In conclusion, I think this manuscript needs to be redone. It cannot be published in its present form.

Author Response

REVIEWER 2

Comments: Thank you for study.

The paper Biomechanics of the upper limbs through wearable sensors: A review in the sports combat ambit, written by authors Andres Blanco Ortega, Jhonatan Isidro Godoy, Dariusz Slawomir Szwedowicz Wasik, Eladio Martinez Rayon, Claudia Cortes Garcia, Hector Ramon Azcaray Rivera, Fabio A. Gomez Becerra is intended as a review of the use of IMU sensors for upper limb movement analysis.

I appreciate that the authors have attempted to make a commendable effort in collecting and analyzing specific material that has appeared over the last 20 years in order to summarize the use of these sensors.

Major observations:

The way the material is presented, however, is more like a course focused on upper limb biomechanics, with theoretical aspects and anatomical images being mentioned, which should already be known by specialists and researchers who would be interested in the article.

Author’s Response: Thank you very much for your comments. The purpose of defining some terms is to put researchers who are beginning in this line of research in context. In addition, this also gives more clarity to the terms used throughout the document. However, taking into account the observation, some figures and concepts were omitted.

Comments: Intended as a review, the material submitted for analysis should follow the PRISMA procedure for analyzing bibliographic references. The manuscript should focus more on the methodology for selecting and choosing the articles, and presenting inclusion and exclusion criteria. The article should present extensively the research and biomechanical measurements selected for this review.

Author’s response: To carry out this research, the guidelines of Preferred Reporting Items for Systematic Reviews and Meta-Analyses (PRISMA) were followed. An information search was carried out in the bases of PubMed, Medline, IEEExplore, Scopus,... The following keywords were used in the search of all the bases: (IMU sensor) AND (upper limb) AND (biomechanics). The selection of the articles was based first on the title, then on the abstract, and finally on the full text. The selection was made by the authors individually, and later meetings were held with all the authors for disagreements.

Comments: Therefore, I believe that chapters III (Spatial Coordinate System), IV (Shoulder Musculoskeletal-Articular System), and Table 1 (Normal angular range of motion in upper limbs), a useful table for a biomechanics course, are not necessary for this type of manuscript.

Author’s Response: Thank you very much for the comments, the recommendation is accepted and major corrections are made to these sections.

Comments: In Chapter V, Methodology, the specific steps of the PRISMA methodology (MDPI recommended methodology for review manuscripts) should be presented.

Author’s Response: Thank you very much for the recommendation, the information is added by applying the PRISMA methodology. It is described in section IV.

Comments: Chapter IX, Mechanical power estimation equations, is interesting from the point of view of mechanical power estimation, but it is not necessary for this type of manuscript, as it has no bibliographic reference and gives a theoretical presentation of how useful muscle energy is divided during the performance of functional movements of the upper limb.

Author’s response: Nosotros creemos que la estimación de la potencia mecánica desde el punto de vista muscular es un parámetro biomecánico de gran importancia. La información esta referenciada en el primer párrafo de la página 12.

Comments: Chapter X, Conclusions, does not sufficiently detail the applications of IMU sensors and the particularities of the use of these sensors in biomechanical assessments, with references to the materials reviewed in this review manuscript.

Some minor criticisms:

The pdf copy of the manuscript does not contain line numbering and for this reason, I cannot make exact references, but the small punctuation errors I noticed will certainly be corrected by the authors when they rewrite the article.

Please note that the title of table 2 is in Spanish.

In conclusion, I think this manuscript needs to be redone. It cannot be published in its present form.

Author’s response: In general, major corrections have been made accepting the suggestions described. Small typographical errors have been corrected, as well as translation errors. Our total thanks and recognition for your invaluable opinion that leads us to be better day by day.

Best regards to all reviewers from the authors.

Reviewer 3 Report

The authors deals with a comprehensive study of wearable devices to get biomechanics about the upper extremities of the human body. The authors use the recent researches from where they extracted the information involving the use of inertial sensors to establish a biomechanical model of human movement. The equation is well known but is also very interesting information for a certain target audience. The article is well written, well understood and can be a theoretical starting point for those who want to develop motion sensors applied to the upper extremities of the human body. From my point of view, this article may be of interest to readers, so I recommend it for publication.

Author Response

REVIEWER 3

Comments: The authors deals with a comprehensive study of wearable devices to get biomechanics about the upper extremities of the human body. The authors use the recent researches from where they extracted the information involving the use of inertial sensors to establish a biomechanical model of human movement. The equation is well known but is also very interesting information for a certain target audience. The article is well written, well understood and can be a theoretical starting point for those who want to develop motion sensors applied to the upper extremities of the human body. From my point of view, this article may be of interest to readers, so I recommend it for publication

Authors Response: It is a pleasure for us to read your comment, we greatly appreciate the time you have spent reviewing our work. We extend our gratitude and hope that this corrected work continues to hold your full approval.

Best regards to all reviewers from the authors.

Reviewer 4 Report

In the review paper, the authors summarized state-of-the-art technologies in measuring biomechanical parameters of upper limbs using wearable sensors. Overall, it is a well-written manuscript with a clear structure. The following amendments are suggested that can further improve the quality of the article.

1: There are minor grammatical errors throughout. It is suggested to thoroughly revise the manuscript and utilize online tools like grammarly.

2: In the abstract, the literature search strategy can be simplified. The detailed conclusions summarized from the literature can be provided.

3. The movement of upper limbs is also be an important factor that influence the accuracy of wearable sensor at different body sites (Refer: 0.3389/fphys.2020.00823, 10.1007/s10877-020-00481-3), which may affect the reliability of monitoring results in clinical application scenarios (Refer: 10.3390/s21020468). This information should be mentioned in the introduction to stress the significance of this review.

4. For figures 3-6, are they from published works? If so, the agreement from copyright owner should be mentioned in the caption.

5. The wireless sensing is another promising technology in detecting body movement (Refer: 10.1002/jnm.2632, 10.3390/app8040568, 10.1109/RADAR41533.2019.171307) of multiple subjects (Refer: 10.3390/s22030809. This technology can be combined with wearable sensors in the future to enhance the reliability and accuracy of biomechanical monitoring of upper limbs. This can be mentioned in the discussion to further improve the quality of the article.

Author Response

REVIEWER 4

Comments: In the review paper, the authors summarized state-of-the-art technologies in measuring biomechanical parameters of upper limbs using wearable sensors. Overall, it is a well-written manuscript with a clear structure. The following amendments are suggested that can further improve the quality of the article.

1: There are minor grammatical errors throughout. It is suggested to thoroughly revise the manuscript and utilize online tools like grammarly.

2: In the abstract, the literature search strategy can be simplified. The detailed conclusions summarized from the literature can be provided.

  1. The movement of upper limbs is also be an important factor that influence the accuracy of wearable sensor at different body sites (Refer: 0.3389/fphys.2020.00823, 10.1007/s10877-020-00481-3), which may affect the reliability of monitoring results in clinical application scenarios (Refer: 10.3390/s21020468). This information should be mentioned in the introduction to stress the significance of this review.

  1. For figures 3-6, are they from published works? If so, the agreement from copyright owner should be mentioned in the caption.

  1. The wireless sensing is another promising technology in detecting body movement (Refer: 10.1002/jnm.2632, 10.3390/app8040568, 10.1109/RADAR41533.2019.171307) of multiple subjects (Refer: 10.3390/s22030809. This technology can be combined with wearable sensors in the future to enhance the reliability and accuracy of biomechanical monitoring of upper limbs. This can be mentioned in the discussion to further improve the quality of the article.

Author’s Response:

1.- The minimal grammatical errors have been identified and evaluated through the Grammarly software and its plug-ins for the text editor program under which this work was written. We extend our gratitude for showing us important tools for correct writing in the English language.

2.- The search strategy was fully detailed in section 4, under the PRISMA methodology, by the recommendations of another reviewer.

3.- The recommendation to mention the work done in the clinical field is accepted. Improving the quality of the writing and referencing the consequences in the rest of the body due to the movement of the upper extremities.

4.- The section containing the figures has been removed, also in consideration by other reviewers. Figure 3 remains due to the importance of knowing the body planes, in addition, we can show it because is designed own.

5.- Radio wave monitoring technology is promising and revolutionary. Suggestion accepted and added to section 6.

Best regards to all reviewers from the authors.

Round 2

Reviewer 1 Report

Some comments from the reviewer have been addressed. Suggest strengthening the discussion on the challenges faced in current research and the perspectives of the future research directions in this field.

Author Response

Reviewer 1:

We are grateful for your comments again. In the last three paragraphs of section 9 'Conclusions',  we have made some modifications (page 15), in response to your suggestions.

We address the difficulties encountered in generating this work, and the pros and cons of using IMU's. Even the new incursions that will be had in the coming investigations.

In addition, future work is mentioned, as well as the expectations that this research group has for the use of this technology in various fields.

Finally, we wish to express our appreciation for your reviews as they have been very helpful, we send you an affectionate greeting. We will be waiting for your evaluation once again.

Reviewer 2 Report

Accept the manuscript in present form. The authors have revised the manuscript in accordance with the comments I made.

Author Response

Reviewer 2:

We are grateful for your final response. We wish to express our appreciation for your reviews as they have been very helpful, we send you an affectionate greeting.

This manuscript is a resubmission of an earlier submission. The following is a list of the peer review reports and author responses from that submission.

Round 1

Reviewer 1 Report

This paper deals with the biomechanics of the upper limbs within the field of sport combat. There is a big gap to be published in Applied Science in the present form. From the reviewer’s point of view, the shortage of this paper relies on these major aspects:

  1. The abstract does not reflect the key points of the manuscript at all, and it needs to be rewritten;
  2. The whole paper is poorly organized;
  3. Apparently, the manuscript focuses on the application of sensors in the biomechanics of the upper limbs, rather than “biomechanics” itself;
  4. The writing of this paper needs to be improved substantially. Lots of grammatical and typing errors.

Reviewer 2 Report

The paper discusses the review of the biomechanics of the upper limb in combat sports. I regret to inform you that the paper does not provide enough information/insights to be considered for publication. Authors claimed in the abstract that they have used only 10 years old references (means no less than 2011) but almost half of them are more than 10 years old, which is very fine for the review article but conflicts with the authors claim. Secondly, the scheme and style of the paper is the major issue. It does not look like paper but some sort of encyclopedia containing a lot of basic definitions and simple figures. Lastly, the paper does not give any comparison between the literature and does not provide any reliable conclusion at the end. 

Reviewer 3 Report

The paper presents a review of the biomechanics of the upper limbs. In total, 28 publications were cited. The papers were obtained  from highly-regarded journals focused on biomechanics. In my opinion, this review has potential, but requires additional work before publication. My main comment is that the manuscript tries to cover too many aspects and most of them are not developed properly, for a review. For instance, section 6 contains a very broad description of the angular displacements in the upper limb and only 2 papers are referenced. I would expect at least a dozen references regarding the range of motion and, for instance, a discussion on how the measurement of ROM can be a complex issue. On the other hand, section 7 is well developed and contains a clear and useful summary of instruments used for biomechanical measurements. Section 9 focuses on general mathematical equations for modeling the motion of body joints. It is interesting, but in my opinion, does not fit within the review. A short part of it could be used in the introduction. In general, I believe that the authors should rethink the main aim of the review and its structure.  As mentioned before, the manuscript has potential, but with its current flow it is difficult to understand its main points and use it properly.   Other minor comments:   1. Language should be revised. The paper contains multiple small spelling errors, which lower the quality of the text. Due to the lack of numbers I cannot enumerate all of them, but some examples in the abstract include: "Inertial sensors [...]. Its miniaturization [...]" "[...] a summary of his use in order to [...]" "[...] human bod"   2. Each figure appears to be drawn in a different style. Nevertheless, the sources for the figures were not included - did you make all the figures? If not, did you obtain permission for republishing the figures (if necessary)?

Reviewer 4 Report

The manuscript entitled “Biomechanics of the upper limbs: A review in the sports combat ambit “ was reviewed as stated further.  This study reports a summary of sports biomechanics, the approach towards this field of study, measuring instruments etc. Unfortunately the manuscript is poorly written, and does not provide a complete discussion on the topic. The subtitles of the manuscript has not been chosen well and the manuscript has no concentration on the topic.

Collection of data was claimed to be from the literature performed in the past 10 years. However, in the bibliography older references were also observed. The manuscript suffers from deprived structural and grammatical English and it certainly needs an English proof reading. This manuscript in the current format is not suitable for publication in the Journal and requires huge changes. My comments on this manuscript are formulated as follows:

Abstract

Apart from typos and grammar issues in the Abstract as shown partly below, it definitely needs revision to better reflect the manuscript.

For instance what do the authors mean by “for obtaining biomechanics” in the last line??

"Inertial sensors over time have become an essential ally for current researchers. Its miniaturization coupled with its ever-increasing reliability make it ideal for certain applications. Therefore, this article will present a summary of its use in order to obtain biomechanical variables such as speed, acceleration and power; with a focus on combat sports involving boxing, karate, tae kwon do among others. A search has been performed through a couple of databases including PubMed, Elseiver, IEEE Publisher, and Taylor and Francis. With this background, research data no older than 10 years have been collected, they are tabulated and classified for the interpretation of reading researchers. Finally, this work provides a wide vision in the use of portable devices for obtaining biomechanics in the upper extremities of the human body"

  1. Introduction

The first two paragraphs attempts to describe what biomechanics is, which is not the interest of readers because probably those referring to this work have already know it. Instead the introduction must discuss more on the studies carried out in upper limb biomechanics and why this review paper is needed and it requires be specific about aspects of this field is covered.

In the 3rd paragraph two sentences are repeated.

  1. Sports applications of Biomechanics.

It is not a convincing subtitle!! There are unnecessary use of „among others“ in the text

Figure 1, the parameter shown on y axis must be clearly specified with the unit (I assume it is time in seconds), but it must be better illustrated.

3. Spatial Coordinate System.

Absolutely unnecessary section for potential readers of this manuscript as these terms are well familiar to biomechanical researchers. There is no referencing for Figure 1-4

5. Methodology.

      The bullet point number 2 stated on page 6 does not fully reflect the title of the manuscript. The authors named the manuscript with somehow a specific title but their search keywords are very general.

  1. Measuring instruments

The way of referring to reference numbers 12, 13, and 14 in the first three  paragraphs of page 8 are inappropriate. The name of author must be shown up.

No reference on Figure 6.

This manuscript contains no novelty and creativity in writing a review paper. It is not specifically written for the title provided. It looks like a student reports it, which adds knowledge only to those who are totally naïve in biomechanics and cannot help the community of professional researchers. This manuscript at the current format is not appropriate for publication. A full revision is required for this.